# Effect of Electroless Cu Plating Ti_3_AlC_2_ Particles on Microstructure and Properties of Gd_2_O_3_/Cu Composites

**DOI:** 10.3390/ma15051846

**Published:** 2022-03-01

**Authors:** Haiyao Cao, Zaiji Zhan, Xiangzhe Lv

**Affiliations:** State Key Laboratory of Metastable Materials Science & Technology, Yanshan University, Qinhuangdao 066004, China; haiyaocao@ysu.edu.cn (H.C.); lvxiangzhe@hotmail.com (X.L.)

**Keywords:** Cu@Ti_3_AlC_2_-Gd_2_O_3_/Cu composites, electroless Cu plating, microstructure, electrical conductivity, mechanical properties

## Abstract

Ti_3_AlC_2_ presents a hexagonal layered crystal structure and bridges the gap between metallic and ceramic properties, and Gadolinia (Gd_2_O_3_) has excellent thermodynamic stability, which make them potentially attractive as dispersive phases for Cu matrix composites. In this paper, Cu@Ti_3_AlC_2_-Gd_2_O_3_/Cu composites, Ti_3_AlC_2_-Gd_2_O_3_/Cu composites, and Gd_2_O_3_/Cu composites were prepared by electroless Cu plating, internal oxidation, and vacuum hot press sintering. The microstructure and the effect of the Cu plating on the properties of the Cu@Ti_3_AlC_2_-Gd_2_O_3_/Cu composites were discussed. The results showed that a Cu plating with a thickness of about 0.67 μm was successfully plated onto the surface of Ti_3_AlC_2_ particles. The ex situ Ti_3_AlC_2_ particles were distributed at the Cu grain boundary, while the in situ Gd_2_O_3_ particles with a grain size of 20 nm were dispersed in the Cu grains. The electroless Cu plating onto the surface of the Ti_3_AlC_2_ particles effectively reduces their surfactivity and improves the surface contacting state between the Cu@Ti_3_AlC_2_ particles and the Cu matrix, and reduces electron scattering, so that the tensile strength reached 378.9 MPa, meanwhile, the electrical conductivity and elongation of the Cu matrix composites was maintained at 93.6 IACS% and 17.6%.

## 1. Introduction

Because of the ease of fabrication, high thermal and electrical conductivities, and excellent corrosion resistance, copper and its alloys are widely used in industrial machinery such as heat exchangers, the divertor components such as fusion reactors, electrical equipment (wiring and motors), and construction for plumbing [1,2,3,4,5]. Particle-reinforced Cu matrix composites strengthened by low-content reinforcement particles could have a high strength performance, but the scattering of electrons was inevitably increased due to the addition of heterogeneous particles in the Cu matrix, consequently reducing the electrical conductivity of the Cu matrix composites. Therefore, the preparation of a particle-reinforced Cu matrix composite with high tensile strength and high electrical conductivity is still a challenge at present.

The reinforced particles can be introduced by in situ synthesizing or ex situ addition into the Cu matrix composites [6,7,8,9,10]. The compressive strength of an in situ (1.0 wt%) Al_2_O_3_/Cu composite synthesized from powders obtained by electrical explosion of wire was increased nearly 1.6 times compared with pure copper, but its electrical conductivity decreased from 95%IACS to 72%IACS [6]. Another Cu matrix composite strengthened by 3.0 wt% ex situ Al_2_O_3_ particles was fabricated using the hot-pressing method [7], the hardness of the composite increased nearly 1.5 times compared with pure copper, but its electrical conductivity was down to 77%IACS. The average size of the in situ and ex situ Al_2_O_3_ particles was 30 nm and 65 μm, respectively. The composite strengthened by ex situ Al_2_O_3_ particles had a weak interface between the Al_2_O_3_ particle and the Cu matrix because of their large misfit in lattice parameters, CTE, and elastic modulus. Cr plating on the surface of the Al_2_O_3_ particles could improve surface contacting state between the reinforced particles and the Cu matrix [8], and the tensile strength of the Cr@Al_2_O_3_/Cu composites was improved comparing with the Al_2_O_3_/Cu composites. In summary, the advantages of the in situ synthesized method over the ex situ method included offering a smaller particle size with the improved surface contacting state between the reinforced phase and Cu matrix, so that a better reinforcement effect could be achieved.

Gadolinia (Gd_2_O_3_) and many other rare earth oxides are more attractive as dispersive phases than Al_2_O_3_ for Cu matrix composites due to their excellent thermodynamic stability [11,12,13,14]. Moreover, those rare earth elements show a low diffusivity and solubility in the metal matrix because of their larger atomic radii, consequently, could improve the microstructural stability of the metal matrix against coarsening [15,16,17,18]. The addition of in situ synthesized Y_2_O_3_ in Cu matrix led to the increase in ultimate tensile strength by nearly three times after cold rolling [11]. Therefore, the Cu matrix composites strengthened by in situ Gd_2_O_3_ nanoparticles were expected to have both high tensile strength and high electrical conductivity.

MAX phases present a hexagonal layered crystal structure and bridge the gap between metallic and ceramic properties [19,20,21,22,23]. Comparing with Al_2_O_3_, TiB_2_, and ZrB_2_, Ti_3_AlC_2_ presents high elastic moduli, high electrical conductivity, and low density relatively [24,25,26,27], which make Ti_3_AlC_2_ a potentially attractive dispersion phase for Cu matrix composites. For example, Cu composite with 60 vol% ex situ Ti_3_AlC_2_ particles prepared by ball milling for 10 h followed by hot press exhibited a superior ultimate compressive strength (1242 ± 24 MPa) and low electrical resistance (0.32 × 10^−6^ Ω·m) [19]. However, due to the limitation of in situ synthesized raw materials, reaction conditions, and reaction time, the content of in situ synthesized reinforcement phase is low, strengthening efficiency of the in situ reinforcement phase was limited, so the combination method of in situ synthesized and ex situ added reinforcement phase provides a way to prepare the particle-reinforced Cu matrix composites towards high tensile strength and high electrical conductivity, which was one innovation point of this paper.

The other innovation point of this paper was to prepare a Cu matrix composite with improved strength and stable electrical conductivity, the approach was to introduce in situ Gd_2_O_3_ (internal oxidation) and ex situ Ti_3_AlC_2_ (directly added) particles into a Cu matrix, and a layer of Cu was coated onto the surface of the Ti_3_AlC_2_ particles before adding them into the Cu matrix to reduce their surfactivity and improve the surface contacting state between the Cu@Ti_3_AlC_2_ particles and the Cu matrix. At the same time, the microstructure evolution and the effect of the Cu plating on the properties of the Cu@Ti_3_AlC_2_-Gd_2_O_3_/Cu composites were also discussed.

## 2. Materials and Methods

### 2.1. Raw Materials and Preparation Procedure

The raw materials for preparing Cu@Ti_3_AlC_2_-Gd_2_O_3_/Cu composites include Cu-Gd alloy powder (Central South University, Changsha, China), cuprous oxide (Cu_2_O) powder, and Ti_3_AlC_2_ powder (Beijing Forsman Technology Co., Ltd, Beijing, China), their purities are above 99.0%, and their sizes are about 10, 2, and 2 µm. The Cu-Gd alloy powder was atomized by high purity Ar gas from the inductive melting of Cu-Gd alloy ingots, which was made from Cu and Gd powders. The element content of Gd in Cu-Gd alloy powder is about 0.84 at% (2.04 wt%), which was analyzed using an X-ray fluorescence spectrometer (ARL, Switzerland).

Figure 1 presents the preparation process diagram of the Cu@Ti_3_AlC_2_-Gd_2_O_3_/Cu composite, which mainly included (a) electroless Cu plating on the surface of the Ti_3_AlC_2_ particles, (b) in situ synthesis of Gd_2_O_3_ reinforced phase, (c) the ball milling and sintering of the Cu@Ti_3_AlC_2_ and Gd_2_O_3_/Cu powders.

#### 2.1.1. Electroless Cu Plating on the Surface of the Ti_3_AlC_2_ Particles

Formaldehyde (HCHO) (Shanghai Aladdin Biochemical Technology Co., Ltd., Shanghai, China) was used as the reductant to coat Cu plating on the surface of Ti_3_AlC_2_ particles in the bluestone solution (CuSO_4_), which could reduce the surfactivity of the Ti_3_AlC_2_ particles, to enhance the surface contacting state between the Ti_3_AlC_2_ particles and the Cu matrix. The reaction of electroless Cu plating on the surface of Ti_3_AlC_2_ particles was followed by the Equation (1):(1)$${\mathrm{Cu}}^{2+}+2\mathrm{HCHO}\;+4{\mathrm{OH}}^{-}\to \mathrm{Cu}↓{+2\mathrm{HCOO}}^{-}{+H}_{2}↑{+2H}_{2}O\;$$

The electroless Cu plating process on the surface of Ti_3_AlC_2_ particles mainly included the following steps: (a) pretreatment before electroless Cu plating, (b) electroless Cu plating process, and (c) treatment after electroless Cu plating.

(a) Pretreatment before electroless Cu plating: firstly, the Ti_3_AlC_2_ particles were cleaned in acetone solution by ultrasonic bath for 10 min to remove the oil and other dirt attached on their surfaces. Then, the Ti_3_AlC_2_ particles were stirred in a solution with a NaOH concentration of 30% and temperature of 363 K for 20 min using a constant temperature electromagnetic stirrer. Finally, the Ti_3_AlC_2_ particles were cleaned with distilled water and dried in a vacuum oven.

(b) Electroless Cu plating process: the Cu electroless plating process was carried out in a constant temperature electromagnetic stirrer (INESA Scientific Instrument Co., Ltd., Shanghai, China) at a temperature of 333~338 K and a pH value of 12.0~12.5 with a stirring speed 60 r/min. Table 1 shows the composition and concentration of the bath solution used in electroless Cu plating on the surface of Ti_3_AlC_2_ powder particles.

The required concentration solution of CuSO_4_·5H_2_O (Shanghai Aladdin Biochemical Technology Co., Ltd., Shanghai, China), 2,2′-bipyridine (Shanghai Aladdin Biochemical Technology Co., Ltd., Shanghai, China), Na_2_EDTA (Shanghai Aladdin Biochemical Technology Co., Ltd., Shanghai, China), and NaOH (Shanghai Aladdin Biochemical Technology Co., Ltd., Shanghai, China) were prepared, respectively, before electroless Cu plating. Then, the prepared Na_2_EDTA solution, the NaOH solution, the 2,2′-bipyridine solution, and the Ti_3_AlC_2_ powder particles were successively and slowly added into the CuSO_4_ solution. During the whole electroless Cu plating process, the electromagnetic stirrer remained running to keep the mixed solution in a uniform condition. The HCHO solution was finally added into the mixture after the temperature reached to 333~338 K and the pH value between 12.0 and 12.5, then the Cu in the mixed solution began to deposit on Ti_3_AlC_2_ particles.

Due to the consumption of OH^-^ ions and the generation of H_2_ during the plating process, the pH value of the mixed solution would decrease, the NaOH solution should be continuously added into the mixed solution to keep the PH value between 12.0 and 12.5. As the electroless plating process went on, the Cu^2+^ in the mixed solution was gradually reduced to Cu and deposited on the surface of Ti_3_AlC_2_ powder particles, the color of the mixed solution gradually became lighter and lighter, and finally colorless and transparent, indicating that almost all the Cu^2+^ in the mixed solution was consumed to Cu to deposit onto the surface of Ti_3_AlC_2_ particles, and the electroless Cu plating process was over.

(c) Treatment after electroless Cu plating: after the reaction, the beaker with the mixed solution was removed from the electromagnetic mixer agitator (INESA Scientific Instrument Co., Ltd., Shanghai, China) and left for stratifying. The size and density of the Ti_3_AlC_2_ particles were greatly increased as a layer of Cu grains was deposited onto the surface of Ti_3_AlC_2_ particles after electroless plating. Therefore, the electroless Cu plating Ti_3_AlC_2_ (Cu@Ti_3_AlC_2_) particles almost all sank at the bottom of the beaker after simple standing for a few minutes.

After pouring out the residual plating solution, distilled water and alcohol were repeatedly used to clean and wash the Cu@Ti_3_AlC_2_ particles alternately three times, and the Cu@Ti_3_AlC_2_ particles were then dried in a vacuum oven.

#### 2.1.2. In Situ Synthesis of Gd_2_O_3_ Reinforced Phase

The Gd_2_O_3_ reinforced phase was in situ synthesized by internal oxidation method according to the following reaction Equation (2) [5]:(2)$$2{\mathrm{Gd}+3\mathrm{Cu}}_{2}{O=\mathrm{Gd}}_{2}O_{3}+6\mathrm{Cu}\;\Delta H_{1198}^{0}=-1329.5\mathrm{kJ}\;\Delta G_{1198}^{0}=-1252.8\;\mathrm{kJ}\;$$

The Cu-Gd alloy powder and Cu_2_O powder were planetary ball milled for 4 h in an Ar atmosphere, the weight ratio of the ball-to-powder and milling speed were 4:1 and 180 rpm, respectively. Then, the mixed powders were oxidized in a vacuum furnace (0.3 Pa) at 1198 K for 1 h. Subsequently, the mixed powders were reduced by high purity hydrogen at 698 K for 2 h to remove excess oxygen, and finally the Gd_2_O_3_/Cu powder was obtained.

#### 2.1.3. The Ball Milling and Sintering of the Cu@Ti_3_AlC_2_ and Gd_2_O_3_/Cu Powders

A certain amount of Cu@Ti_3_AlC_2_ powder and Gd_2_O_3_/Cu powder were planetary ball milled under the same milling condition in Section 2.1.2. Then, the mixed powders containing 1.5 wt% Ti_3_AlC_2_ and 2.0 wt% Gd_2_O_3_ were prepressed at 10 MPa, followed by sintering in a vacuum hot pressure sintering furnace (Shanghai Chenhua Science Technology Co., Ltd., Shanghai, China) (3.0 × 10^−2^ Pa) at 1173 K for 30 min. The uniaxial pressure of 30 MPa and heating rate of 10 K/min were applied. Ti_3_AlC_2_-Gd_2_O_3_/Cu composites (1.5 wt% Ti_3_AlC_2_ without electroless Cu plating and 2.0 wt% in situ Gd_2_O_3_) and Gd_2_O_3_/Cu composites (2.0 wt% in situ Gd_2_O_3_) were also prepared under the same conditions for comparison, a certain amount of spherical pure Cu powder (99.9% purity, ∼10 µm in diameter) was added to maintain the same amount of reinforcement in the composites.

### 2.2. Test Methods

The morphologies and microstructures of the reinforced particles and the interfaces in the composites were conducted by scanning electron microscopy with an acceleration voltage of 25 kV (SEM, KYKY-EM3200, KYKY, Beijing, China) and transmission electron microscopy at 300 kV (TEM, Tecnai G2 F30, FEI, Hillsboro, OR, USA), which was equipped with an energy-dispersive X-ray spectrometric (EDS, Mahwah, NJ, USA) system. The ex situ Ti_3_AlC_2_ powder, ex situ Cu@Ti_3_AlC_2_ powder, the mixed powders of Cu-Gd alloy powder and Cu_2_O powder before internal oxidation and after H_2_ reduction were evaluated by X-ray diffraction (XRD, D/MAX-2500/PC, Tokyo, Japan) method, respectively. The electrical conductivity of the samples was tested with a four probes method at ambient temperature, the tip distance of the four probes is 1 mm, and the size of the electrical conductivity sample is φ 30 mm × 4.5 mm. Nine different areas in each sample were randomly selected for the electrical conductivity tests to minimize the error. A TH5000 universal testing machine (Jiangsu Tianhui Experimental Machinery Co., Ltd., Yangzhou, China) was used for tensile tests at ambient temperature according to GB/T 228.1-2010, the crosshead speed was 0.3 mm/min. The schematic diagram of the tensile specimens is shown in Figure 2, the gage length was 10 mm, and cross sections of 2 × 1.5 mm, respectively. Three specimens were tested from each sintered billet for tensile and electrical conductivity tests to minimize the error. The average tensile strengths and elongation of the samples were calculated from the recorded tensile stress–strain curves.

## 3. Results and Discussion

### 3.1. Electroless Cu Plating on the Surface of Ti_3_AlC_2_ Particles

The morphologies and XRD patterns of the Ti_3_AlC_2_ powder and Cu@Ti_3_AlC_2_ powder were conducted to analyze the quality of the Cu plating. Figure 3a shows the angular morphology of the Ti_3_AlC_2_ particles with a size of about 2 μm, the Ti_3_AlC_2_ particles became smooth and ellipsoidal after electroless Cu plating (Figure 3b), and fine size Cu particles were uniformly distributed onto the Ti_3_AlC_2_ particles’ surfaces without micropores. From Figure 3c,d, Ti_3_AlC_2_ particle was completely wrapped by the Cu plating without gaps or micropores, and the thickness of the Cu plating varied from 0.36 to 0.98 μm and averaged 0.67 μm. Figure 3e presents the XRD patterns of the Ti_3_AlC_2_ powder and Cu@Ti_3_AlC_2_ powder. Strong peaks of Ti_3_AlC_2_ were seen in Ti_3_AlC_2_ powder in Figure 3e; the characteristic peak of Cu was sharp and strong, while the characteristic peak of Ti_3_AlC_2_ was weak in Cu@Ti_3_AlC_2_ powder in Figure 3e (Ti_3_AlC_2_:PDF#52-0875, Cu: PDF#04-0836). The initial weight of Ti_3_AlC_2_ powder was 0.50 g, which increased to 4.11 g (Cu@Ti_3_AlC_2_) after electroless copper plating. The calculated average thickness of the Cu plating of 0.66 μm was obtained according to the theoretical density of Ti_3_AlC_2_ and pure Cu, the weight increment, and the size of Ti_3_AlC_2_; the value was close to the average value in Figure 3d.

### 3.2. In Situ Synthesis of Gd_2_O_3_

Figure 4 shows the powder analysis of in situ synthesis of Gd_2_O_3_, and Figure 4a presents SEM images of Cu-Gd alloy powder mixed with Cu_2_O powder before internal oxidation. The diffusing distance of O in the Cu_2_O into the Cu-Gd alloy powder was reduced because the fine Cu_2_O powder adhered to the surface of the larger Cu-Gd alloy powder after ball milling, which was beneficial to the reaction between the Gd in Cu-Gd alloy powder and O in Cu_2_O powder during the internal oxidation process, and the synthesis of Gd_2_O_3_ would be easier and more efficient. Figure 4b shows the SEM images of Cu-Gd alloy powder mixed with Cu_2_O powder after H_2_ reduction, the Cu_2_O powder adhering to the surface of Cu-Gd alloy powder gradually integrated after the mixed powder was oxidized at 1198 K for 1h and reduced by H_2_ at 698K for 2 h, meanwhile, the surface of the mixed powders became rougher. Figure 4c presents the XRD patterns of the mixed powders in Figure 4a,b, a weak characteristic peak of Gd_2_O_3_ (222) could be seen in the XRD patterns of the mixed powders after internal oxidation and H_2_ reduction, indicating that Gd_2_O_3_ reinforced phase was in situ synthesized. At the same time, the characteristic peak of Cu_2_O disappeared, indicating that Cu_2_O was reacted to Cu during the H_2_ reduction.

### 3.3. Microstructure

#### 3.3.1. The Distribution of Ti_3_AlC_2_ in the Cu Matrix

In order to analyze the effect of electroless Cu plating on the Ti_3_AlC_2_ particles and the distribution of the Ti_3_AlC_2_ phase in the Cu matrix, the samples of the Ti_3_AlC_2_-Gd_2_O_3_/Cu composite and the Cu@Ti_3_AlC_2_-Gd_2_O_3_/Cu composite were inlaid in one billet for polishing. SEM and EDS images in both composites were collected and their results are shown in Figure 5.

The dark particles with a size of about 2μm were uniformly distributed in the light-colored area, as shown in Figure 5a,b. From Figure 5a, small cracks and micropores (indicated by red arrows) existed between the dark particles and the matrix in the Ti_3_AlC_2_-Gd_2_O_3_/Cu composite, while the dark particles were closely combined with the matrix without gaps and micropores in the Cu@Ti_3_AlC_2_-Gd_2_O_3_/Cu composite (Figure 5b). Figure 5c shows that the elements in area A in Figure 5b were Ti, Al, C, and Cu, and their atomic percentages were 45.65%, 17.05%, 36.27%, and 1.03%, respectively. The proportion of Cu element was very small, and the ratio of Ti, Al, and C elements was close to 3:1:2, so the dark particle at area A was Ti_3_AlC_2_. All elements in area B in Figure 5b were Cu, indicating that the light-colored area was the Cu matrix.

From Figure 5a, cracks in Ti_3_AlC_2_-Gd_2_O_3_/Cu composites were mainly seen in the interface between the Ti_3_AlC_2_ particles and the Cu matrix due to their large misfit in lattice parameters, CTE, and elastic modulus. By electroless plating Cu onto the surface of the Ti_3_AlC_2_ particles, the surface contacting state between the Ti_3_AlC_2_ particles and Cu matrix was improved and no cracks at the interface were observed in the Cu@Ti_3_AlC_2_-Gd_2_O_3_/Cu composites.

#### 3.3.2. Microstructure of the Gd_2_O_3_/Cu Composites

Figure 6 shows the TEM and EDS images of Gd_2_O_3_/Cu composite prepared by the internal oxidation method, which was used for further confirming the components of the particles dispersed in the matrix. Figure 6a displays the bright field image, and dark particles (marked as A) and a large bright flat area (marked as B) were selected for EDS analysis. The bright field image showed that the spherical reinforcement phase with a size of about 20 μm was distributed in the matrix. Only Gd, O, and Cu elements existed in the dark area A from the EDS spectrum in Figure 6c, and the atomic percentages of Gd and O elements were 39.53% and 57.30%, their atomic ratio was close to 3:2, illustrating that the spherical reinforcement phase was Gd_2_O_3_, which was synthesized in the composite during the internal oxidation process. Only Cu existed in area B (Figure 6d), which indicated that the bright area was the Cu matrix.

Further analysis of the high-resolution TEM image of area A in Figure 6a showed that the interplanar spacings of 0.3118 and 0.1802 nm matched with the Gd_2_O_3_ (222) plane (0.3122 nm, PDF 12-0797) and Cu (200) plane (0.1808 nm, PDF 04-0836), respectively (Figure 6b). The Gd_2_O_3_ (222) in Figure 6 was consistent with the XRD result in Figure 4c. The Gd_2_O_3_ particles existed in the Cu matrix without other phases between them due to the in situ nucleation and growth of the Gd_2_O_3_ particles in the Cu matrix, which improved their interface strength and the tensile strength of the Gd_2_O_3_/Cu composite.

#### 3.3.3. Microstructure of the Ti_3_AlC_2_-Gd_2_O_3_/Cu Composites

Figure 7a shows the bright field image of the Ti_3_AlC_2_-Gd_2_O_3_/Cu composite, and Figure 7b shows the high-resolution image of the white box in Figure 7a. According to the calibration analysis of the high-resolution image, the interplanar spacings of d_1_ and d_2_ were 0.2118 and 0.1845 nm, which were highly in accordance with Ti_3_AlC_2_ (105) plane (0.2159 nm, PDF#52-0875) and Cu (200) plane (0.1808 nm, PDF#04-0836), respectively, because the percentage of the differentials between the test values and the standard values is less than 2%.

There was a micropore between the Cu matrix and the Ti_3_AlC_2_ particles (the yellow oval in Figure 7a), which might be caused by the high surface energy between the Ti_3_AlC_2_ particles and the Cu matrix due to their structural characteristics, resulting in the defect during the sintering process. The reinforced phase Ti_3_AlC_2_ had a clear boundary with the Cu matrix, which was mechanically bound. This indicated that the interfacial bonding between the Cu matrix and the Ti_3_AlC_2_ particles without Cu plating was weaker, which may weaken the enhancement effect. The result further confirmed the finding in Section 3.3.1.

#### 3.3.4. Microstructure of the Cu@Ti_3_AlC_2_-Gd_2_O_3_/Cu Composites

In order to assess the effect of the electroless Cu plating on the Ti_3_AlC_2_ particles, the TEM images of the Cu@Ti_3_AlC_2_-Gd_2_O_3_/Cu composite were collected. Figure 8 presents the TEM and EDS images of Cu@Ti_3_AlC_2_-Gd_2_O_3_/Cu composite prepared by electroless Cu plating, internal oxidation, and vacuum hot press sintering.

The fine dark particles were dispersed in the Cu matrix (highlighted by yellow arrows in Figure 8a), which should be the in situ Gd_2_O_3_ inferred by their sizes in Figure 6a. In order to confirm that the dark particles were the in situ Gd_2_O_3_, the high-resolution TEM image was taken and analyzed as shown in Figure 8c, the interplanar spacings of 0.3151 and 0.2091 nm were in accordance with the Gd_2_O_3_ (222) plane and Cu (111) plane, respectively. The in situ Gd_2_O_3_ particles were in situ nucleated and grew up in the Cu matrix without other phases between them, which improved the tensile strength of the composite. The larger white particles were distributed at the Cu grain boundary in Figure 8a, the EDS (Figure 8b) showed that the main elements in area A were Ti, Al, and C, and only 1.58% of Cu existed, the atomic ratio of Ti, Al, and C elements was close to 3:1:2; the result suggested that the white phase should be Ti_3_AlC_2_. The high-resolution TEM image at area C was analyzed to further confirm the components. As shown in Figure 8d, the interplanar spacings were 0.2186 and 0.1831 nm and in accordance with the Ti_3_AlC_2_ (105) plane and Cu (200) plane, respectively. Results indicated that the white enhanced phase at area C was Ti_3_AlC_2_. There were no micropores and cracks between the Ti_3_AlC_2_ particles and the Cu matrix because of the electroless Cu plating on the surface of the Ti_3_AlC_2_ particles.

The in situ Gd_2_O_3_ particles and the Ti_3_AlC_2_ particles in the Cu matrix acted as obstacles to the moving dislocations according to the Orowan strengthening mechanism [28,29], and the dislocation density around the reinforcements in the Cu matrix might be increased due to the thermal expansion coefficient difference between the Cu matrix and reinforcing phase [30,31], leading to an increase in the strength of the Cu@Ti_3_AlC_2_-Gd_2_O_3_/Cu composite.

### 3.4. Tensile Strength and Electrical Conductivity

Figure 9 shows the stress–strain curve (a), tensile strength and electrical conductivity (b) of three different composites: Gd_2_O_3_/Cu composite, the Ti_3_AlC_2_-Gd_2_O_3_/Cu composite, and the Cu@Ti_3_AlC_2_-Gd_2_O_3_/Cu composite.

As could be seen from Figure 9a, a large plastic deformation occurred before fracture during the tensile test, the tensile strength was improved by ex situ Ti_3_AlC_2_ particles from 325.4 (Gd_2_O_3_/Cu composite) to 351.4 MPa (Ti_3_AlC_2_-Gd_2_O_3_/Cu composite), the modulus of the Ti_3_AlC_2_-Gd_2_O_3_/Cu composite was also increased at the same time, but the elongation decreased from 26.1 ± 0.2% to 15.2 ± 0.3%. After electroless Cu plating on the surface of the Ti_3_AlC_2_ particles, the tensile strength of the Cu@Ti_3_AlC_2_-Gd_2_O_3_/Cu composite was further increased to 378.9 MP and elongation at break reached 17.6 ± 0.2%. The tensile strength and elongations of the Cu@Ti_3_AlC_2_-Gd_2_O_3_/Cu composite were improved by 27.5MPa and 2.4%, respectively, comparing with the Ti_3_AlC_2_-Gd_2_O_3_/Cu composite. The electrical conductivity decreased slightly from 95.0%IACS of the Gd_2_O_3_/Cu composite to 92.9%IACS of Ti_3_AlC_2_-Gd_2_O_3_/Cu composite by directly adding Ti_3_AlC_2_ particles, but it still remained above 90.0%IACS. Comparing with the Ti_3_AlC_2_-Gd_2_O_3_/Cu composite without electroless Cu plating on the surface of the Ti_3_AlC_2_, the electrical conductivity of the Cu@Ti_3_AlC_2_-Gd_2_O_3_/Cu composite with Cu plating was slightly increased to 93.6%IACS. The electrical conductivity and elongation of the Cu@Ti_3_AlC_2_-Gd_2_O_3_/Cu composites prepared by the combination of in situ synthesized and ex situ added methods were 93.6%IACS and 17.6% (this paper), which were much higher than that of the TiB_2_/Cu composite in [9]. However, the tensile strength of the Cu@Ti_3_AlC_2_-Gd_2_O_3_/Cu composite was lower than that of the TiB_2_/Cu composite because 1 wt% Fe was added into the TiB_2_/Cu composite.

Comparing with the Gd_2_O_3_/Cu composite, the tensile strength increase in the Ti_3_AlC_2_-Gd_2_O_3_/Cu composite was due to the pining of the dislocation movement caused by the in situ Gd_2_O_3_ particles in the Cu grains and the Ti_3_AlC_2_ particles at the boundary of the Cu grains according to the grain boundary strengthening, Orowan strengthening, thermal mismatch strengthening, and load transfer strengthening mechanisms [29,32,33,34]. The decrease in electrical conductivity in the Ti_3_AlC_2_-Gd_2_O_3_/Cu composite was due to the electron concentration decrease and the increases in free electron scattering caused by the Ti_3_AlC_2_ particles at the Cu grain boundary [35]. Comparing with the Ti_3_AlC_2_-Gd_2_O_3_/Cu composite, the tensile strength and electrical conductivity of the Cu@Ti_3_AlC_2_-Gd_2_O_3_/Cu composite were simultaneously increased because the surfactivity between the Cu matrix and the Ti_3_AlC_2_ particles was reduced after electroless Cu plating on their surface, which led to the better surface contacting state and less electron scattering after sintering.

For the Ti_3_AlC_2_-Gd_2_O_3_/Cu composite, the load would be transferred from the Cu matrix to the Ti_3_AlC_2_ particles during the tensile test, which hindered the plastic deformation of the Cu matrix and reduced its elongation. With the addition of Cu@Ti_3_AlC_2_ particles into the Gd_2_O_3_/Cu composite, the interfacial bonding between the Cu@Ti_3_AlC_2_ particles and Cu matrix was improved due to the Cu plating on the surface of the Ti_3_AlC_2_ particles. The opportunity of crack initiation between the Cu@Ti_3_AlC_2_ particles and Cu matrix was low under the same load condition, resulting in the continuous plastic deformation of the Cu matrix, so increased tensile strength and elongation of Cu@Ti_3_AlC_2_-Gd_2_O_3_/Cu composite were seen.

### 3.5. Fracture Morphology

A large number of dimples with uniform sizes distributed on the fracture surface of the Gd_2_O_3_/Cu composites are shown in Figure 10a, and these typical ductile fractures indicated that the Gd_2_O_3_/Cu composite was plastically deformed before fracture. The enlarged image in Figure 10b showed in situ Gd_2_O_3_ particles existing at the bottom of the dimples, which proved the movement of the dislocation would be impeded by the in situ Gd_2_O_3_ particles during the tensile test and helped to improve the tensile strength of the Gd_2_O_3_/Cu composite [26,27]. The size of the in situ Gd_2_O_3_ particles was about 20 nm, and the value was consistent with the results in Figure 6a,b and Figure 8c.

There were relatively obvious gaps between the Cu matrix and the Ti_3_AlC_2_ particles in the fracture morphology of the Ti_3_AlC_2_-Gd_2_O_3_/Cu composite (Figure 10c,d), and Ti_3_AlC_2_ particles with a size of about 2 μm existed at the bottom of the dimples. After electroless Cu plating on the surface of the Ti_3_AlC_2_ particles, no gaps existed in the Cu@Ti_3_AlC_2_-Gd_2_O_3_/Cu composite, as shown in Figure 10e,f, and it could be seen that dimples with uniform size were distributed in the fracture morphology of the Cu@Ti_3_AlC_2_-Gd_2_O_3_/Cu composite.

The reason for the gaps formed in the Ti_3_AlC_2_-Gd_2_O_3_/Cu composite might be the weak interface bonding due to their large misfit in lattice parameters, CTE, and elastic modulus between the Ti_3_AlC_2_ particles and the Cu matrix. During the tensile testing process, the load would be transferred from the Cu matrix to Ti_3_AlC_2_ particles via the interface; with the increase in the load, the Ti_3_AlC_2_ particles could hinder the dislocation movement so that the tensile strength of the Ti_3_AlC_2_-Gd_2_O_3_/Cu composite was improved. As the load increased further to the level that exceeded the interface bonding strength, the cracks initiated from the interface between Ti_3_AlC_2_ particles and Cu matrix and interface debonding happened, so the gaps between Ti_3_AlC_2_ particles and Cu matrix were observed. With the electroless Cu plating onto the surface of the Ti_3_AlC_2_ particles, the surfactivity between the Cu@Ti_3_AlC_2_ particles and the Cu matrix was reduced, and a better surface contacting state was achieved after the sintering process, so that no interface debonding was formed between the Cu@Ti_3_AlC_2_ particles and the Cu matrix during the tensile process. Higher tensile strength and elongation was achieved in the Cu@Ti_3_AlC_2_-Gd_2_O_3_/Cu composite.

## 4. Conclusions

According to the preparation and performance analysis of the Cu@Ti_3_AlC_2_-Gd_2_O_3_/Cu composites, Ti_3_AlC_2_-Gd_2_O_3_/Cu composites, and Gd_2_O_3_/Cu composites, the innovation points of the combination method and the composites with high tensile strength and electrical conductivity were achieved, and the major findings were as follows:

The Cu@Ti_3_AlC_2_-Gd_2_O_3_/Cu composites with high tensile strength and electrical conductivity were prepared using the combination method of in situ synthesized and ex situ added reinforcement phase.

The Gd_2_O_3_ particles were in situ synthesized during the internal oxidation, and the Gd_2_O_3_ particles with a size of about 20 nm were dispersed in the Cu grains. The Ti_3_AlC_2_ particles were successfully wrapped by a Cu plating without gaps and micropores, which existed at the Cu grain boundary.

The interfacial bonding between the Ti_3_AlC_2_ particles and Cu matrix was effectively improved by the Cu plating onto the surface of Ti_3_AlC_2_ particles, and the tensile strength, reached 378.9 MPa, meanwhile, electrical conductivity and elongation of the Cu@Ti_3_AlC_2_-Gd_2_O_3_/Cu composites remained at 93.6 IACS% and 17.6%.

Only the reinforcement by in situ rare earth oxides and ex situ MAX phases were characterized in this paper, and reinforcements using other particles in Cu matrix composites will be studied in the future.

## Figures and Tables

**Figure 1 materials-15-01846-f001:**
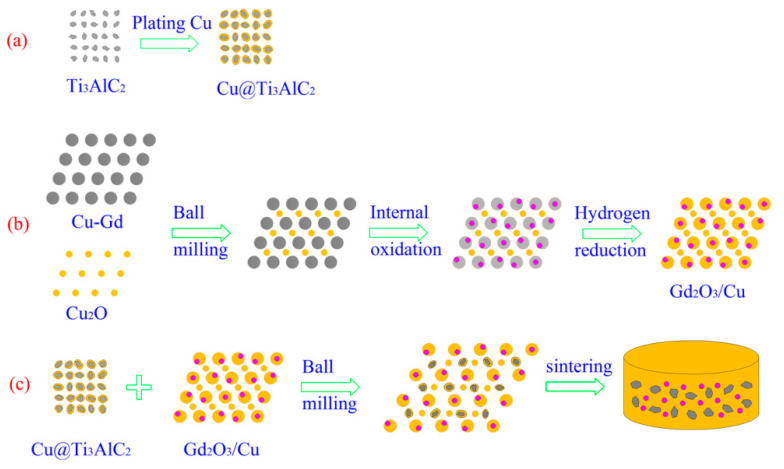
The preparation process diagram of the Cu@Ti_3_AlC_2_-Gd_2_O_3_/Cu composite: (**a**) electroless Cu plating on the surface of the Ti_3_AlC_2_ particles, (**b**) in situ synthesis of Gd_2_O_3_ reinforced phase, (**c**) the ball milling and sintering of the Cu@Ti_3_AlC_2_ and Gd_2_O_3_/Cu powders.

**Figure 2 materials-15-01846-f002:**
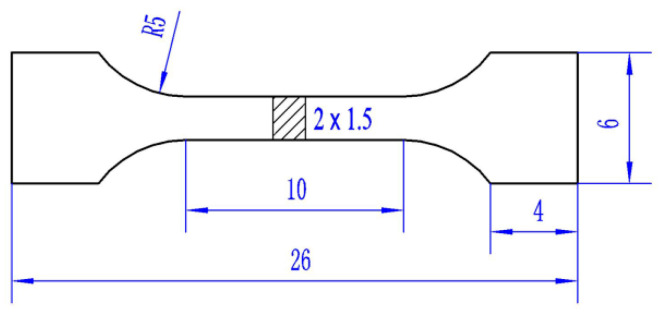
Schematic diagram of tensile specimen (unit: mm).

**Figure 3 materials-15-01846-f003:**
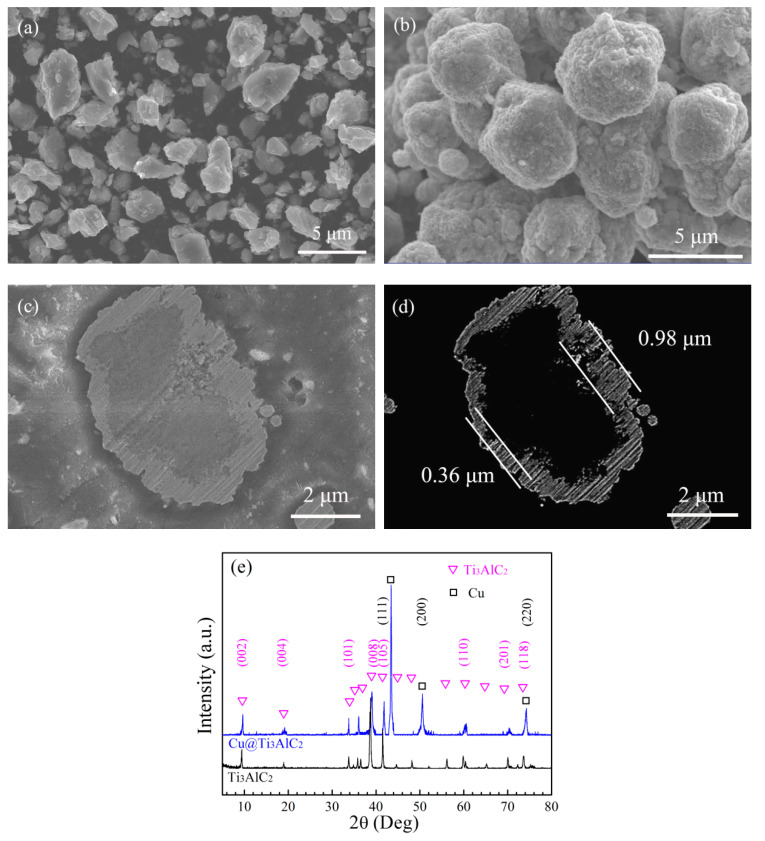
The Ti_3_AlC_2_ powder particles before and after electroless Cu plating: (**a**,**b**) the morphology of Ti_3_AlC_2_ powder and Cu@Ti_3_AlC_2_ powder; (**c**,**d**) the profile of the Cu@Ti_3_AlC_2_ powder; (**e**) the XRD patterns of Ti_3_AlC_2_ powder and Cu@Ti_3_AlC_2_ powder.

**Figure 4 materials-15-01846-f004:**
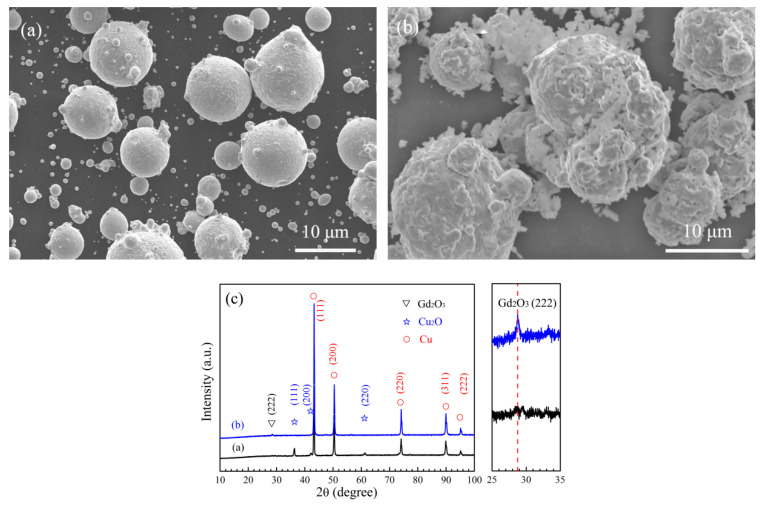
The powder analysis of in situ synthesis of Gd_2_O_3_: (**a**,**b**) the morphology of Cu-Gd alloy powder mixed with Cu_2_O powder before internal oxidation and after H_2_ reduction; (**c**) the XRD patterns of the mixed powders in Figure 4a,b.

**Figure 5 materials-15-01846-f005:**
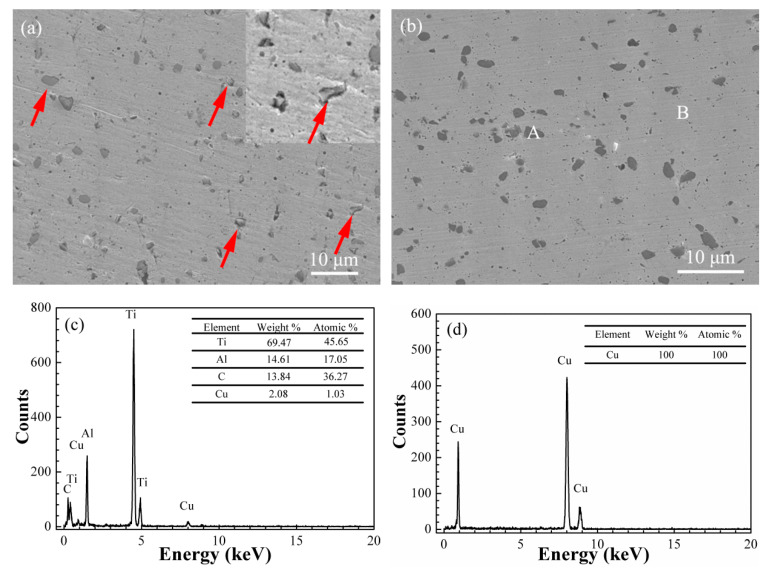
The distribution of the Ti_3_AlC_2_ particles in the Cu matrix: (**a**) Ti_3_AlC_2_-Gd_2_O_3_/Cu; (**b**) Cu@Ti_3_AlC_2_-Gd_2_O_3_/Cu; (**c**,**d**) are the energy spectra of areas A and B in Figure 5b; small cracks and micropores were indicated by red arrows.

**Figure 6 materials-15-01846-f006:**
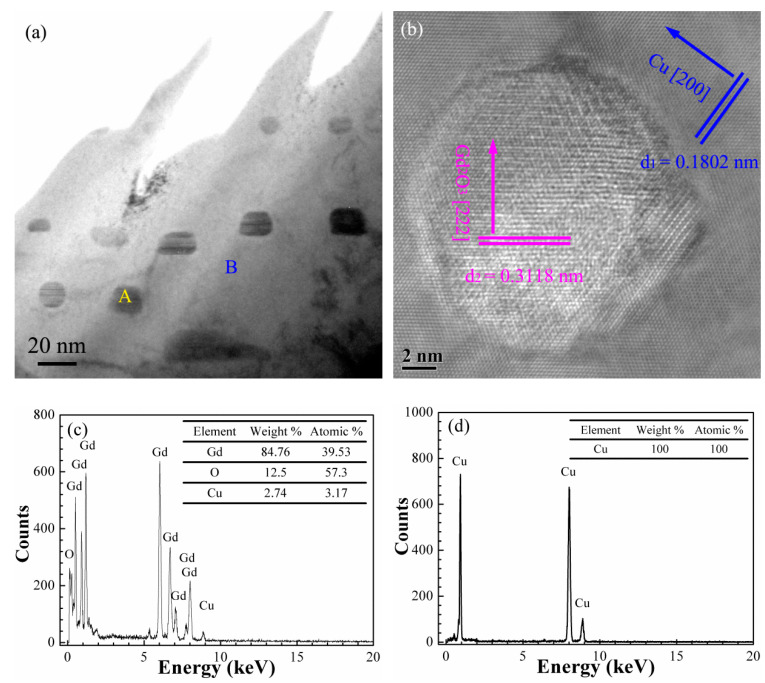
The TEM and EDS images of the Gd_2_O_3_/Cu composite: (**a**) bright field image; (**b**) high resolution image; (**c**,**d**) the corresponding EDS spectrum at areas A and B in Figure 6a respectively.

**Figure 7 materials-15-01846-f007:**
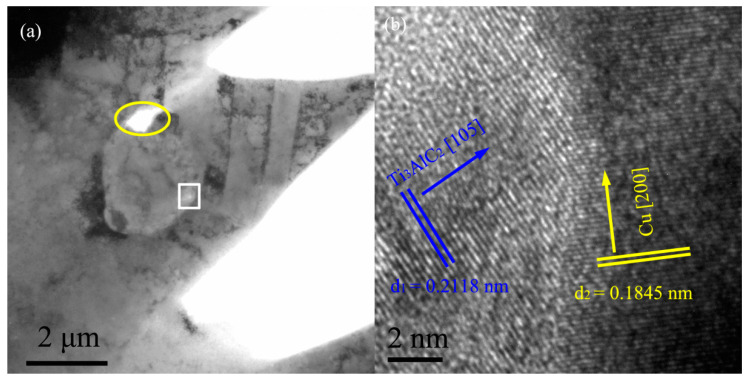
The TEM images of the Ti_3_AlC_2_-Gd_2_O_3_/Cu composite: (**a**) bright field image; (**b**) high resolution image; a micropore was indicated by the yellow circle and the high resolution image was from the white box in Figure 7a.

**Figure 8 materials-15-01846-f008:**
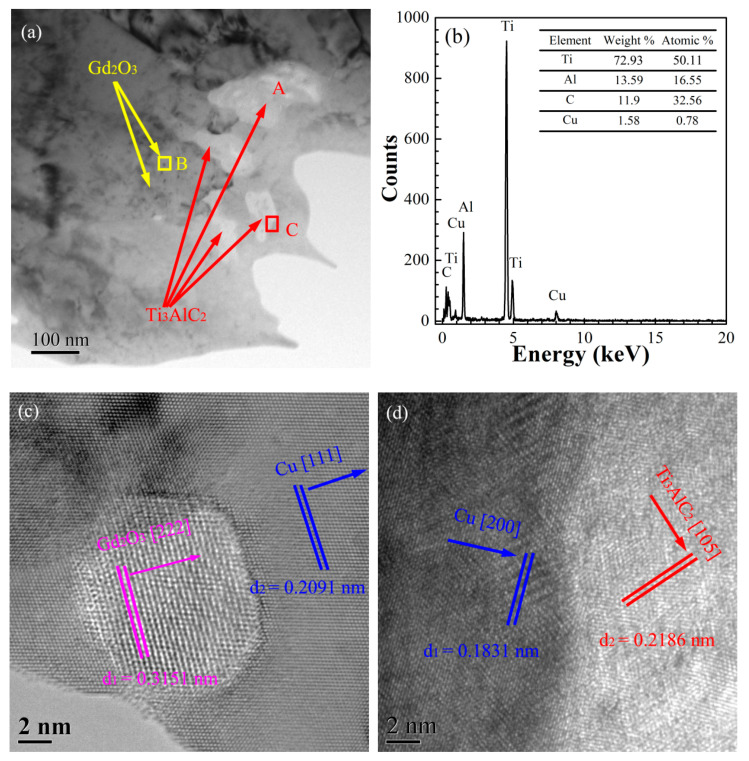
The TEM and EDS images of Cu@Ti_3_AlC_2_-Gd_2_O_3_/Cu composite: (**a**) the bright field image; (**b**) the energy spectra of area A in Figure 8a; (**c**,**d**) are the high-resolution images of areas B and C in Figure 8a, respectively.

**Figure 9 materials-15-01846-f009:**
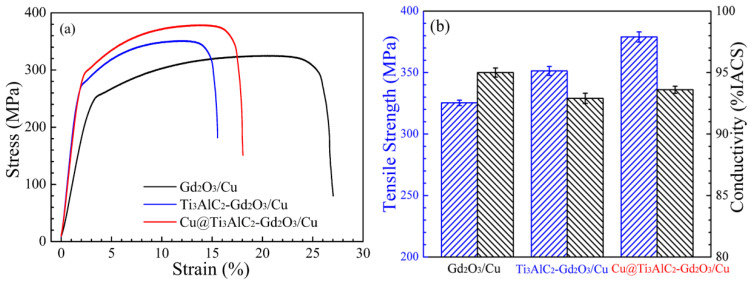
The stress–strain curve, tensile strength, and electrical conductivity of the composites: (**a**) the stress–strain curve and (**b**) the tensile strength and electrical conductivity.

**Figure 10 materials-15-01846-f010:**
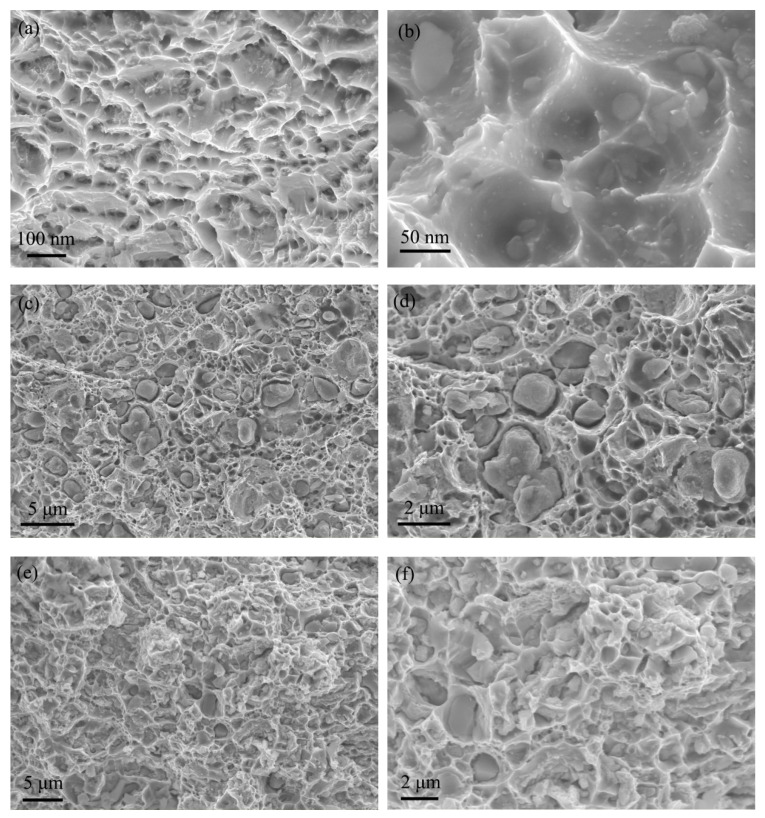
The fracture morphology of the composites: (**a**,**b**) the Gd_2_O_3_/Cu composite; (**c**,**d**) the Ti_3_AlC_2_-Gd_2_O_3_/Cu composite; (**e**,**f**) the Cu@Ti_3_AlC_2_-Gd_2_O_3_/Cu composite.

**Table 1 materials-15-01846-t001:** The composition and concentration of bath solution.

Function	Main Salt	Reductant	PH Regulator	Stabilizer	Complex
Composition	CuSO_4_·5H_2_O	HCHO (37%)	NaOH	2,2′-Bipyridyl	Na_2_EDTA
Concentration	18 g/L	20 mL/L	8 g/L	0.05 g/L	30 g/L

## Data Availability

Data is contained within the article.

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
