# Peer review of "Effect of Electroless Cu Plating Ti3AlC2 Particles on Microstructure and Properties of Gd2O3/Cu Composites"

_materials, 2022, doi:10.3390/ma15051846_

Round 1

Reviewer 1 Report

The manuscript entitled "Effect of electroless Cu plating Ti3AlC2 particles on micro-structure and properties of Gd2O3/Cu composites” prepared different composites by electroless Cu plating, internal oxidation, and vacuum hot press sintering. The microstructure and the effect of the Cu plating on the properties of these composites were discussed.

The manuscript is good in quality and well written. I suggest acceptance after considering the following comments.

Comments:

  • A similar check shows 31% similarity. It should be reduced in the revised manuscript.
  • The authors should increase their discussion on previous related research and highlight how their study is providing a different approach or adding significantly to what has been done.
  • Section 2.2: Sketches including dimensions in lines 167-168 and showing the details of the tested specimens should be provided. Moreover, the test setup should be provided in the revised manuscript.
  • Line 173: I think it should be "shows", not "showed". It should be simple present and should be modified throughout the manuscript.
  • Line 264: I think the word "respectively" should be provided. the same note is for Figure 7.
  • Lines 290-298: 105, 222, 111, and 200: I cannot see these reference numbers at the end of the manuscript. The authors should correct this error throughout the manuscript.
  • Section 3.4: The authors should highlight the effects on the modes of failure, ductility, and stiffness of the new material. What are the improvements in the mechanical properties? This should be highlighted in the conclusion section.
  • Line 318: It will be better to add how much improvement in the tensile strength and elongations.

Reviewer 2 Report

- Improve the introduction with data about Cu matrix composites from the literature.

- Add Miller indices on XRD patterns.

- Highlight better which is the novelty of the work?

- What is the status of the literature according to your work? Make a comparison between the results obtained by you and another previous research.

- Add more conclusions. Complete the conclusions with the limitations of the proposed methodology. Also write future research.

- Generally, the quality of the writing could be improved.

Reviewer 3 Report

Interesting study to combine in-situ and ex-situ methods for high-strength and high-conductivity Cu MMNCs. However, the authors ignored the state-of-art literatures, provided insufficient discussion into the electrical conductivity change, and lacked experimental support/information, which makes the current manuscript shaky and not too convincing.

I would suggest a major revision.

  1. The authors have ignored many exciting improvements for Cu nanocomposites. The references appear too old. I would suggest add more related up-to-date literatures like below (particularly my colleague Prof. Xiaochun Li, Prof. Soon Hyung Hong, and Prof. Daolun Chen’s group has investigated this area for years):

[1] Pan, S., Zheng, T., Yao, G., Chi, Y., De Rosa, I. and Li, X., 2022. High-strength and high-conductivity in situ Cu–TiB2 nanocomposites. Materials Science and Engineering: A, 831, p.141952.

[2] Hwang, J., Yoon, T., Jin, S.H., Lee, J., Kim, T.S., Hong, S.H. and Jeon, S., 2013. Enhanced mechanical properties of graphene/copper nanocomposites using a molecular‐level mixing process. Advanced materials, 25(46), pp.6724-6729.

Otherwise, I do not see the difference and improvements (by property comparison) from the current research.

  1. For Cu-Gd alloy powder, what is the element content? Better add Cu-Gd phase diagram.
  2. Equation (1): Electrons are not conserved…H and O atoms are not conserved!!! This is a serious problem!
  3. For 4-probe test, what is the sample shape? Film? Bulk? What is the thickness?
  4. Acceleration voltage for SEM and EDS?
  5. If Cu can wrap Ti3AlC3 so well (Fig. 2d), why not directly add Ti3AlC3 to Cu? Do you have TEM images to show that Ti3AlC3 could be wrapped well (only) by electroless plating? What difference could Cu-wrapping do basically?
  6. Figure 2e, give the standard card indices. (ICPDS index, etc.) For Cu@Ti3AlC2, it seems tht you did background removal? What is the fitting background? Fig.2 and Fig 3 XRD should have the same processing.
  7. Is the interface between Gd2O3 and Cu coherent or incoherent? What is the interface influence on mechanical and electrical property?
  8. For electrical conductivity and tensile properties, how many times of tests have you done? What is the stdev for ductility?
  9. Based on Figure 9d, I do not think the wrapping of Cu onto Ti3AlC2 is effective and provides good interface…All the Ti3AlC2 interface is already pulled out… The authors clearly need more experimental support… Otherwise I would doubt their properties.
  10. Interface is also for Cu electrical conductivity, but this discussion is lacked in the current manuscript. Please look into the following references for a DEEPER discussion about electron behavior and electrical conductivity in MMNCs…

[1] Pan, S., Yuan, J., Zhang, P., Sokoluk, M., Yao, G. and Li, X., 2020. Effect of electron concentration on electrical conductivity in in situ Al-TiB2 nanocomposites. Applied Physics Letters, 116(1), p.014102.

Reviewer 4 Report

I appreciate the authors contribution on providing a new way of reinforcing Cu matrix. The paper contains interesting scientific and experimental findings. Unfortunately, the paper lets one understanding that a single trial on the topic was performed, hence I consider that more repeated trials would much better support the paper topic.

Aiming to improve the paper, I have commented some statements you posted in the paper as follows.

Line 44> “The composite strengthened by ex-situ Al2O3 particles had a weak interface between the Al2O3 particle 45 and the Cu matrix due to their poor surface wettability. Cr plating on the surface of the 46 Al2O3 particles could improve the surface wettability between the reinforced particles 47 and the Cu matrix [8]”

Comment 1. The author use wettability in a confusing manner i.e. wettability can be used when Cu matrix is melted, but not when it is solid.

See wikipedia: Wetting is the ability of a liquid to maintain contact with a solid surface, resulting from intermolecular interactions when the two are brought together. This happens in presence of a gaseous phase or another liquid phase not miscible with the first one. The degree of wetting (wettability) is determined by a force balance between adhesive and cohesive forces.

https://en.wikipedia.org/wiki/Wetting#cite_note-1:

Comment 2. What means P in Eq. (1) and where does it disappears?

Line 80 > The raw materials were Cu-Gd alloy powder (99.9% purity, ∼10 μm in diameter),

Comment 3. The composition of the Cu-Gd alloy is unspecified. Is it unimportant?  

Line 149-157> “A certain amount of Cu@Ti3AlC2 powders and Gd2O3/Cu powders were planetary ball milled at a speed of 180 rpm with a ball-to-powder weight ratio of 4:1 for 4 h in an Ar atmosphere. Then the powders containing 1.5 wt% Ti3AlC2 and 2.0 wt% Gd2O3 were firstly compacted at 10 MPa, and sintered in a vacuum hot pressure sintering furnace (3.0 × 10-2 Pa) under a uniaxial pressure of 30 MPa. During this sintering process, the furnace was heated up at a speed of 10 K/min, the sintering temperature and holding time were 1173 K and 30 min. Ti3AlC2-Gd2O3/Cu composites (1.5 wt% Ti3AlC2 without electroless Cu plating and 2.0 wt% Gd2O3) and Gd2O3/Cu composites (2.0 wt% Gd2O3) were also prepared under the same conditions for comparison.”

Comment 4: The Cu matrix seems being missed in aforementioned experiments!  Please explain?

Line 162 “equipped with an energy-dispersive X-ray spectroscopy...system”!

Comment 5: I suggest spectrometric instead spectrometry

Comment 5: Nothing about XRD in $2.2., but XRD outcomes are posted in the paper.  

Line 182 :„the thickness of the Cu plating varied with an average thickness 178 of 0.67 μm.”

Comment 6: The above statement is feeble if it is based only on 2 measurements carried on a single particle. Many more particles should be measured to get a statistics!!!

Line 182-184 „In addition, no other peak appeared in Figure 2e, which indicated that the Cu plating had no other 183 impurity phase or no oxidation on their surfaces.”

Comment 7: The above statement is doubtful because XRD has a detection limits around 3-5 %v for the crystalline phases and 100 %v for amorphous ones. Hence, Cu@Ti3AlC2 powder may be contaminated at a certain level.

Comment 8: The usage of „powders” for powder seems being wrong some times. It should be checked!.

Lines 198: „Figure 3b was the morphology of Cu-Gd alloy powders mixed with Cu2O powders ..”

Line 2002> „Figure 3c was the XRD patterns of the mixed powders...”

Comment 9:  The above expressions should be checked!

Line 229: „From Figure 4a, cracks in Ti3AlC2-Gd2O3/Cu composites were more likely to initiate 229 in the interface between the Ti3AlC2 particles and the Cu matrix due to their poor wetta-230 bility. By electroless plating Cu onto the surface of the Ti3AlC2 particles, the wettability 231 between the Ti3AlC2 particles and Cu matrix improved, no cracks at the interface were 232 seen so that a better interfacial bonding achieved in the Cu@Ti3AlC2-Gd2O3/Cu compo-233 sites.”

Comment 10: Please explain why did you used wettability, if the composites were obtained by sintering solid components, and no one of them had passed through a melt state during sintering process.

Line 244: „The bright field image showed that the spherical reinforcement phase  with an average size of 20 μm was distributed in the matrix.”

Comment 11: The statement “spherical reinforcement phase with an average size of 20 μm” is not sustained by enough TEM outcomes. Is there supplementary data supporting this statement?

Line 291: The in-situ Gd2O3 particles were metallurgically bonded with the Cu matrix 291 without other phases, which improved their interface strength.

Comment 12: Please explain the meaning of “metallurgicaly bonding”

Line 269 .0.2118 nm and 0.1845 nm, which were in accordance with Ti3AlC2 [105] plane (0.2159 nm, PDF#52-0875) and Cu [200] plane (0.1808 nm, PDF#04-0836), respectively.

Line 322.  The electrical conductivity decreased slightly from 95.0 %IACS of 322 the Gd2O3/Cu composite to 92.9 %IACS

Comment 13 : Some comparisons are risky, as above, if the uncertainties of the compared values are not known.

Line 359. The average size of the in-situ Gd2O3 particles was 20 nm, and the value was consistent with the results in Figure 5a, 5b and Figure 7c.

Comment 14: Hard to say!  Ibid lines 363-367!

Round 2

Reviewer 2 Report

The work has been significantly improved, I suggest checking the work again, there are some more mistakes (eg Line 26, word "Besause" needs to be corrected with "Because").

Reviewer 3 Report

The authors have made significant revision to the manuscript.

Hope the future work of the authors could really keep updated with those leading groups and their publications in the fields.

Reviewer 4 Report

I appreciate that the authors have provided responses to the all my comments. The issues regarding the Comments no. 2-11 were corrected as I have suggested. On the other hand,  I consider that the main shortcomings were avoided through tangled explanations as follows.

 Response 1: ...... The wettability in this paper was used for improving the interfacial bonding between the Cu@Ti3AlC2 particles and the Cu matrix during the sintering process. A Cu plating with fine grains was wrapped on the surface Ti3AlC2 particles after the electroless Cu plating, which reduced the surface energy of the Ti3AlC2 particles. The diffusion ability between the Cu@Ti3AlC2 particles and the Cu matrix (interdiffusion between the Cu plating and Cu matrix) was better than that between the Ti3AlC2 particles and the Cu matrix, which might be caused by the high surface energy between the Ti3AlC2 particles and the Cu matrix due to their structural characteristics.

Comment 1. Irrelevant answer. The wettability term would never improve the interfacial bonding between the Cu@Ti3AlC2 particles and the Cu matrix during the sintering process.

It looks like the authors have tried to bury their mistake in long confusing explanation, avoiding recognise it!

Response 12: Metallurgicaly bonding in Line 291 means that the in-situ Gd2O3 particles were bonded with the Cu matrix by their atoms forces due to their diffusion. The Gd2O3 particles were in-situ synthesized in the Cu-Gd alloy during the internal oxidation process, in which O element from Cu2O diffused into the Cu-Gd alloy and reacted with Gd to form Gd2O3.

Comment 2. There are some types of inter-atomic bonds: metallic, covalent, ionic, van der Waals, but no metallurgical one!

The paper still may confuse the reader through introducing a new type of inter-atomic bonding i.e. metallurgicaly bonding.

Response 13: The comparisons in Line 269 are risky, if the uncertainties of the compared values are not known. But we know that the reinforced particles in this paper are Gd2O3 particles and the Ti3AlC2 particles, which were confirmed by the EDS, XRD and TEM results

Comment 3. The main question is: how much are the uncertainties of the posted data?

For, example, a value of  0.2118 nm implies an exactness of 0. 001 Å close to the amplitude of lattice vibration. Is this true or a higher uncertainty may exists?

Have you been  aware of such an exactness which implies the freezing  of the electron waves and of the lattice vibration ?

The  ISO 98-3:2010 specifies that the numerical values of a measurand cannot be compared unless their uncertainties are assessed. Though, you compared numerical data in your article, finding some improvements.

Comment 4. The response letter did not clarify if a single trial was performed on the topic of the paper or many. It seems to me that this answer has been intentionally omitted, though it is a very important issue. Therefore, I consider that the posted data are of lower reliability as no reproducibility is reported.
